# Outcomes of Consumer Involvement in Mental Health Nursing Education: An Integrative Review

**DOI:** 10.3390/ijerph17186756

**Published:** 2020-09-16

**Authors:** Kyung Im Kang, Jaewon Joung

**Affiliations:** 1Department of Nursing, Dongguk University, Gyeongju 38066, Korea; fattokki@gmail.com; 2Department of Nursing, Semyung University, Jecheon 27136, Korea

**Keywords:** consumer involvement, mental health nursing, nursing education, review

## Abstract

This integrative review analyzed the research on consumer involvement in mental health nursing education in the last decade. We aimed to derive the main contents, methods, and outcomes of education using consumer involvement for mental health nursing students. We searched six electronic databases using English and Korean search terms; two authors independently reviewed the 14 studies that met the selection criteria. Studies on the topic were concentrated in Australia and some European countries; most of them used a qualitative design. The main education subject was recovery, and consumers tended to actively participate in education planning. Moreover, students’ perceptions about education using consumer involvement and people with mental health problems changed positively, as well as their experiences of participating in mental health nursing education. There is a lack of interest in the topic in Asian countries, including Korea. Thus, future studies in Asian countries are needed to conduct qualitative and in-depth explorations of students’ experiences regarding an educational intervention that uses consumer involvement as a tool rigorously designed for mental health nursing education. Consumer involvement can be an innovative strategy to produce high-quality mental health nurses by minimizing the gap between theory and practice in the undergraduate program.

## 1. Introduction

The lifetime prevalence of mental illness in South Korea is 25.4% [1] and, in the United States, it ranges from 23.5% in the Asian population to 45.6% in the White population [2]. Therefore, continuous attention and improvement are required in the mental health care system, policies, and community mental health projects. Moreover, since mental health nurses play a key role in this field, it is important to provide integrated mental health nursing education in the undergraduate program to discharge the students as competent mental health nurses after graduation.

However, the public’s negative perception and attitude toward mental illness are well known, and nursing students share them too [3]. These negative attitudes result in low preferences and intentions toward the mental health nursing field as a future occupation [4,5,6], which may consequently reduce the quality of mental health nursing care. Moreover, since no differences in attitude were found among students before and after completing a formal psychiatric nursing theory course [3], or based on clinical practice [5], an innovative mental health nursing education method should be considered.

It is noteworthy that a consumer is a person who is currently using or has used a service. Consumers of mental health services, as either inpatients or outpatients, are considered patients [7]. Consumer involvement in mental health nursing education allows students to gain valuable insights from the consumer’s perspective and help provide user-centered care [8]. This tool is actively implemented mainly in the UK, Australia, and Northern Europe, but is rare throughout Asia, including Korea. Given that nursing education institutions and consumers operate differently regarding consumer involvement by country [8], it is necessary to establish some foundational data about consumer involvement in mental health training for nursing students by integrating the results of successful research from various countries. In particular, analyzing the outcomes of this type of education from the nursing students’ perspectives rather than from teachers’/educational providers’ perspectives may help in the development of a more effective educational plan.

## 2. Background

Consumer involvement in education is used in medicine, pharmacy, and social welfare science [9,10,11]. In nursing education, this concept was introduced through the specific subfield of physical examination education during the 1960s [12]. In the 1990s, different studies were conducted on consumer involvement in the general nursing field [13]. The utilization of the consumer involvement concept in the mental health nursing education field started in Melbourne, Australia, in 2001 [14], and it is now the most developed field regarding consumer involvement in the context of education [15].

Patients are in the best position to provide a genuine understanding of the mental health service usage, thus helping students overcome stigma by learning about patients’ “genuine” experience with this type of service directly from the source [16]. Nursing students are sometimes criticized for perceiving and interacting with consumers as cases in a textbook, rather than treating them as unique individuals, as a result of current nursing education, which focuses on imparting expertise and knowledge [17]; namely, there is a predominant focus on the perception of patients as disease cases and a concomitant lack of information on how to provide humanized care for these patients. Moreover, consumer involvement in education gives students a deeper understanding of the patient’s experience, and experiences with real patients enable them to improve their confidence and build up reflective knowledge [18]. Besides, the experience of meeting with the consumer (i.e., mental health patient) in the lecture is qualitatively different from that in the clinical practice [19]; furthermore, the story of the consumer is integrated into the curriculum in the context of evidence-based expertise, not just as an ad-hoc story about the use of mental illness and mental health services [20].

Despite these advantages, there is a limit to the practical use of consumers in mental health nursing education compared to the policy emphasis on education using consumer involvement [21]. Previous studies have shown that the subthemes of consumer experience utilization implemented in nursing education differ by country [22]. Given that the use of consumer experience in mental health nursing education reflects socio-cultural differences, there is a need to implement and evaluate education using consumer involvement in various countries. Furthermore, for a seamless global application, it is necessary to secure an appropriate framework for developing educational programs for mental health nursing. Therefore, this study intends to identify the latest trends in mental health nursing education using consumer involvement and provide evidence to develop educational programs that can be utilized in the future. We comprehensively reviewed and analyzed the literature related to mental health nursing education using consumer involvement in undergraduate courses.

## 3. Research Method

### 3.1. Research Aim and Design

This integrated literature review examines the attributes and outcomes of mental health nursing education using consumer involvement as applied to nursing students by analyzing its use in Korea and abroad. We followed Whittemore and Knafl’s [23] five steps for an integrated literature review: (1) problem identification, (2) literature search, (3) data evaluation, (4) data analysis, and (5) result presentation. In particular, at the data analysis stage, we thoroughly followed the data removal, data display, data comparison, conclusion drawing, and verification steps suggested by Whittemore and Knafl [23] for integrated results.

### 3.2. Research Questions

The research questions of this study are as follows:–What were the attributes of consumer involvement in mental health nursing education in the last 10 years?–What were the outcomes of consumer involvement in mental health nursing education for nursing students in the last 10 years?

### 3.3. Literature Search

Before the literature search, the following specific selection and exclusion criteria were prepared. The inclusion criteria for the papers in the literature review were: (1) Papers subjected to a peer-review process before being published in academic journals from 2011 to 2020, (2) papers related to the outcomes of consumer involvement in mental health nursing education that was provided to undergraduate nursing students, and (3) papers written in Korean or English. The exclusion criteria were: (1) Papers including participants other than consumers/patients (e.g., caregivers, service providers, or public involvement) and (2) papers other than original research papers (e.g., dissertations, editors’ letters, conference presentation papers, and review papers, etc.).

The literature search was conducted from 1 April 2020 to 1 June 2020, by two independent researchers, targeting papers related to consumer involvement in mental health nursing education published in domestic and foreign academic journals from 2011 to 2020. The scope of mental health nursing education was based on the regular curriculums conducted at the college level for undergraduate students. Domestic search engines used were RISS (i.e., Research Information Sharing Service), which is the media through which the Korea Education and Research Information Service provides domestic and foreign academic information, and KISS (i.e., Korean studies Information Service System). Foreign search engines used were PubMed, CINAHL (Cumulative Index to Nursing and Allied Health Literature), EMBASE, and PsycInfo. The Medical Subject Headings (MeSH) were used to select search terms, and the English titles and keywords of the papers we found were reviewed to identify high-frequency keywords.

Consequently, search formulas of ((mental health nursing) OR (psychiatric nursing)), ((expert by experience) OR (service user) OR (user involvement) OR (consumer participation)), and ((education) OR (training)) were established. Next, literature retrieved from the database was cataloged through a bibliographic management program (EndNote X9), reviewed, and organized. First, the title, abstract, and full text of the documents were screened step by step—according to the inclusion and exclusion criteria after removing duplicate documents—and the final literature to be analyzed was selected through a discussion among the researchers. No papers were found in the domestic database, and after duplicates were removed from the 1736 papers found in the foreign databases, a total of 1386 documents were extracted. After screening the titles and abstracts, 1372 documents were excluded, totalling 14 documents selected for the final analysis (Figure 1). The entire process of study selection was conducted by two independent researchers and, in case of disagreement, the researchers discussed until they reached a consensus.

### 3.4. Quality Evaluation of Literature

According to Whittemore and Knafl [23], there is no optimized standard for conducting integrated literature reviews. The authors defined this type of research as the process of integrating research papers of various research designs; therefore, it may not be appropriate to use a single quality evaluation tool. Moreover, the authors enforce the importance of the strict application of selection and exclusion criteria when selecting the papers to be analyzed [23]. Therefore, we evaluated the quality of the papers using the Matrix Method [24], and selected the 14 final documents by the precise application of the selection criteria.

The matrix was prepared by dividing the 14 papers into the following sections: Authors, publication year, research design, and main content (i.e., research target information, education subject, research tools, and research results). Two co-researchers evaluated the quality of the papers, determining that all were relevant.

### 3.5. Literature Analysis

The matrix used in the quality evaluation process was used as the framework for the literature analysis. The general characteristics and methods of the selected studies were analyzed in the order of the country of study, the research design, and the number of samples, whereas the characteristics and outcomes of education using consumer involvement were analyzed in the order of the education provision method, the duration of education, and the educational outcomes. The analysis of the literature was conducted by two researchers, who independently reviewed and integrated 14 documents; discussion and reexamination were repeated until a common opinion was drawn for inconsistent evaluations.

## 4. Results

### 4.1. General Characteristics of the Literature

All 14 analyzed papers came from outside Korea; five papers were from Australia (35.7%), three from the UK (21.4%), and the rest were multinational joint research projects, including countries such as Iceland, Finland, Ireland, Norway, the Netherlands, and Australia. Most papers were published in 2019 with five papers (35.8%), followed by three in 2013 (21.4%), and two in 2014 (14.4%). One paper was identified for each of the other years. Ten papers (71.4%) used a qualitative design based on thematic analysis and a phenomenological approach using individual or focus group interviews. The four quantitative studies used questionnaires; their educational outcomes were identified through a pre–post evaluation, and they utilized either a single group or a cohort design. The distribution of the number of participants was 12 people in 5 papers and 51 in 5 papers for qualitative research, and 68 to 194 for quantitative research (Table 1).

### 4.2. Topics of Education Using Consumer Involvement

A total of 11 studies addressed the issue of recovery. Specific topics included the principles and factors related to recovery, the role of nurses in recovery, the formation of cooperative relationships with participants, and the combination of experience and academic factors related to recovery. Additionally, one paper focused on the development of interpersonal skills [25] and another dealt with disease-related symptoms, treatment, and patients’ experiences with nursing care of mentally ill patients from their diagnosis to their recovery [7]. The last study did not mention any specific topics [15].

### 4.3. Education Delivery for the Consumer

In mental health nursing education, consumer participation was conducted face-to-face. Education delivery methods included lectures, providing feedback on learning activities (verbal and written), acting as a facilitator of learning (five papers), or participating in education preparation (nine papers); the latter covered all stages of mental health nursing education (Table 2). Only three studies described the processes and methods of consumer involvement. In the study by Byrne et al. [26], the consumer participated in the process of education delivery, assessment, and even evaluation with equal autonomy to nursing educators. In a study by Stacey et al. [27], consumers were educated on the philosophy and practice of inquiry-based learning and went through class preparation steps that were carefully designed and applied by educational providers, including setting agreements on the specific boundaries of questioning and self-exposure. Additionally, the story of each person was used to create a trigger for learning, and, after communicating with the student in a simulated scenario, verbal or written feedback about the experience was provided by consumers. A study by Stacey and Pearson [25] found that mental health social enterprises, not schools, were responsible for recruiting and preparing the consumers for involvement in mental health nursing education programs. In addition, six papers, from 2019 to 2020, referred to the collaboration and application of a consumer involvement educational model conducted through a multinational project called COMMUNE (Co-production of Mental Health Nursing Education). In these papers, consumer involvement was referred to as “Experts by Experience (EBE)-led teaching.” Although the description of the specific steps regarding how consumers got involved in the process was insufficient, the guidelines provided by the project showed that consumers were involved in various steps, ranging from work preparation to direct education delivery [28].

### 4.4. The Outcomes of Mental Health Nursing Education Using Consumer Involvement

The quantitative and qualitative outcomes of mental health nursing education using consumer involvement could be divided into the following four categories. First, there were changes in students’ perceptions toward education using consumer involvement [7,16,29,30]. Students reported a positive change in their perceptions of consumers’ abilities and their roles as educational participants in an educational model that uses consumer involvement. Consumers facilitated a new level of understanding for the students and created the learning process in which the students engaged, improving the quality of mental health nursing education. The education provided many opportunities to learn about mental health nursing directly from consumers who have lived experience of mental illness.

Second, there were changes regarding nursing students’ perceptions toward people with mental health problems [16,30,31,32], including their beliefs and attitudes toward mental health nursing or patients, their perspectives toward patients’ recovery, and their understanding of patients (i.e., seeing them through a new dimension); also, there were positive changes in their intentions toward and focus on mental health care and a reduction of negative stereotypes regarding this field. The consumer-involved education, which reflected consumers’ lives in a broader context, beyond their diagnosis, facilitated the reexamination of undergraduate students’ preconceptions and untested ideas about mental illness.

Third, nursing students reported changes and new experiences regarding mental health nursing education compared to how they experienced the traditional education on the topic [16,25,27,30,32,33,34]. Education using consumer involvement led to changes in students’ approaches to practice, their self-awareness, reflection, and empathy; it also promoted a more in-depth understanding of the learning contents by encouraging them to recall their past experiences with their families and acquaintances and relate to what they discussed in their classes, thinking about the mental-health-related problems of those around them. Furthermore, active learning was induced by raising questions about students’ principles regarding mental health. In addition, education using consumer involvement provided an opportunity for students to improve their nursing ability, expand their scope of understanding about mental health nursing, and acknowledge the value of this field. However, some students experienced negative changes, such as an excessive anxiety or discomfort caused by the desire to be recognized by a facilitator (consumer) and hide personal challenges or incompetence during class.

**Table 2 ijerph-17-06756-t002:** Summary of consumer/patient involvement in educational programs regarding psychiatric and mental health nursing.

Authors (Year)	Participants	Education Subject	Consumer Involvement	Significant Outcomes
Happell et al. [7]	68	Lived experiences of patients with a major psychotic illness (i.e., diagnosis, treatment, recovery model, nursing care, attitudes as a mental health nurse)	Two-hour lecture	Measuring the attitude of nursing students toward consumer involvement by a consumer participation questionnaire: – A significant improvement in the “consumer as staff” scale (t = −2.07, *p* = 0.04).
Byrne et al. [26]	12	Recovery for mental health nursing practice	Lectures on lived experience, providing advice and support in a non-directive manner when discussing experiences after role play, autonomous coordination, and teaching;	Themes on experiences with consumer involvement: – “Recovery—bringing Holistic Nursing to Life”; – “Influencing practice”; – “Gaining self-awareness through course assessment: Challenge and opportunity
Byrne et al. [16]	12	Recovery for mental health nursing practice	Lectures on lived experiences, autonomous coordination (e.g., education delivery, assessment, evaluation), and teaching	Themes on nursing students’ views of and experiences with consumer involvement: – “Looking through fresh eyes: What it means to have a mental illness”; – “It’s all about the teaching”.
O’Donnell and Gormley [15]	12 (2 focus groups)	Not specified	Not specified	Student perceptions about consumer involvement: – Assessments about consumer involvement: ⋅ Perceptions: Consumer involvement can promote learning and professional development; ⋅ How should consumer involvement be planned and operationalized: Early start or sequenced/layer approach; ⋅ The degree: It can be utilized from simple consumer feedback to formal consumer involvement; ⋅ Perception of formal consumer involvement: Most are welcome, but some are apprehensive. – Adding value to mental health services: ⋅ Equality, partnership, and fairness. – How to protect service consumers during their involvement in nursing education: ⋅ Vulnerabilities of consumers, nursing students, and nurse teachers should be analyzed and understood; ⋅ There is a need for appropriate and comprehensive preparation before consumer involvement.
Byrne et al. [29]	110, consumer-led course 61, nurse-led course	Recovery in mental health nursing	The lived experience-led course	Comparison of between-group differences using the “Mental Health Consumer Participation Questionnaire:” – Consumer-led group: Consumer capacity (t = −3.63, *p* < 0.005) and consumer as staff (t = −5.63, *p* < 0.005); – Nurse-led group: Consumer involvement (t = −3.40, *p* = 0.001) consumer as staff (t = −4.12, *p* < 0.005).
Happell et al. [31]	131, consumer-led course 70, nurse-led course	Recovery approach to care	Autonomous coordination (e.g., coordinate the course, content, delivery)	Measuring nursing students’ attitudes toward people with mental illness: – Consumer-led group: Future career (t = −2.74, *p* = 0.007), preparedness for the mental health field (t = −3.24, *p* < 0.001), and negative stereotypes (t = −3.28, *p* < 0.001); – Nurse-led group: Valuable contributions (t = −5.08, *p* < 0.001) and preparedness for the mental health field (t = −7.83, *p* < 0.001).
Stacey et al. [27]	112, first-year nursing students receiving inquiry-based learning	Mental health recovery	Co-facilitator of inquiry-based learning; Utilizing elements of their personal story as triggers for learning	Themes on the lived experiences of co-facilitators: – Assimilation of new understandings: ⋅ Change in preconceptions; ⋅ Strength and resilience to achieve a better quality of life. – Understanding that previously acquired information was irrelevant to or disconnected from the reality of mental health patients; – Questioning of prior understanding or theoretical principles about mental health; – Discomfort within the learning environment: ⋅ Uncomfortable and triggering for the personal event (e.g., family mental health issues).
Stacey and Pearson [25]	15, final year nursing students	Interpersonal skill assessment	Verbal (15-min) and written feedback on students’ initial interview (30-min) in a simulated scenario	Themes on the nature of learning based on the feedback given by consumers: – Positive outcomes: Increased self-awareness and empathy; – Consideration: Occurrence of personal anxiety among nursing students.
Happell et al. [30]	194	Mental health recovery (COMMUNE project)	Not specified	Self-report measures: The Mental Health Nurse Education Survey (MHNES), The Health Care version of the Opening Minds Scale (OMS), The Consumer Participation Questionnaire (CPQ): – Scales related to attitudinal changes with statistical significance: ⋅ Australia: Social distancing (OMS), Preparedness for the mental health field (MHNES); ⋅ Ireland: Negative stereotypes (MHNES), Preparedness for the mental health field (MHNES), Consumer involvement (CPQ); ⋅ Finland: Negative stereotypes (MHNES), Social distancing (OMS), Preparedness for the mental health field (MHNES), Consumer involvement (CPQ), Lack of capacity (CPQ), Consumer as staff (CPQ).
Happell et al. [33]	51 (8 focus groups)	Mental health recovery (COMMUNE project)	EBE-led teaching	– Main theme: Changing the mindset; – Subthemes: ⋅ Exposing stereotype (i.e., nursing students changed their initial stereotypical views of people who use mental health services); ⋅ Becoming more reflective (i.e., nursing students became more critical and reflective in their approach to mental health nursing).
Happell et al. [32]	51 (8 focus groups)	Mental health recovery (COMMUNE project)	EBE-led teaching	– Main theme: Understanding the person behind the diagnosis; – Subthemes: ⋅ Person-centered care/seeing the whole person being treated; ⋅ Getting to know, understanding, and listening to the person; ⋅ Challenging the medical model, embracing recovery.
Happell et al. [34]	51 (8 focus groups)	Mental health recovery (COMMUNE project)	EBE-led teaching	– Main theme: Improvement in the understanding of mental health nursing; – Subthemes: ⋅ Mental health is everywhere; ⋅ Becoming better nurse practitioners. – Appreciating mental health nursing.
Happell et al. [22]	51 (8 focus groups)	Mental health recovery (COMMUNE project)	EBE-led teaching	– Main theme 1: Getting the structure right; – Subthemes 1: ⋅ Extending the EBE content: There should be more classes and ongoing involvement; ⋅ Best positioning of the EBE content: At the beginning of the program; ⋅ Assessment requirements detract from EBE content: There should be greater consistency between aspects of EBE learning and those in the course; particularly, aspects of the EBE content should be incorporated into assessments; ⋅ More consistency to produce better outcomes: Integrating scientific nursing and EBE knowledge throughout the program. – Main theme 2: Changes to content and approach; – Subthemes 2: ⋅ Allowing multiple perspectives (exchanging between EBE and academic approaches). – Balancing the positive and the negative aspects of both approaches;
Happell et al. [20]	51 (8 focus groups)	Mental health recovery (COMMUNE project)	EBE-led teaching	– Main theme: Bridging the theory and practice gap through first-hand experiences; – Subthemes: ⋅ Bridging theory and life: Deepened understanding of the relevance of theoretical materials to clinical practice; Broadened understanding that goes beyond the clinical perspective; ⋅ Cannot be taught any other way: The unique perspectives EBE brought to the education environment could not be taught by any other methods. – Innovative teaching methods fueling interest and curiosity: The teaching techniques were more innovative compared to what students experienced in other units throughout the nursing course; additionally, EBE-led teaching fueled their curiosity and encouraged them to gain insight into patients’/consumers’ lives.

EBE = Expert by Experience, COMMUNE = Co-production of Mental Health Nursing Education.

Fourth, students reported that their experiences with and assessments of education using consumer involvement [15,20,22] were changed after its application. This type of education helped students to acknowledge mental health services as being characterized by equality, partnership, and fairness; they realized that these services could serve as learning triggers, which provided them with opportunities for professional development by an in-depth understanding that could not be taught by any other methods. However, this type of education had some flaws. The implementation could be better planned; there should be a preparation stage comprising education from nurse academics and consumers; the educational programs should be organized through sequenced or layered approaches, and there should be changes to the educational structure and to the contents of the course that use consumer involvement. Specifically, if the educational content is presented in the context of the unique experiences of individual consumers [32], or if the subjective (e.g., judging the entire mental health care system based on personal experiences, past treatments that no longer reflect the reality of the mental health service) and negative perceptions of the consumer about mental health services (e.g., drug treatment) are reflected in the class content [22], this type of content should be accompanied by objective and balanced coordination, which should be provided before the presentation of consumer-related content (e.g., nurse academics should endeavor to encourage students to reflect on the differences in experience between students and consumers or provide information regarding any issues that may arise during consumer involvement processes). It was also pointed out that consumer involvement should be more clearly integrated into the current/traditional educational methods [22].

## 5. Discussion

This study was conducted to provide a useful framework for utilization in the future development of educational interventions that use consumer involvement. We aimed to do this by identifying the contents, methods, and outcomes of this type of education when being applied to nursing students in the mental health nursing context for the past 10 years.

The literature review revealed that studies on education using consumer involvement are very rare in Korea; they concentrate in some foreign countries (mainly Australia) and qualitative research methods were dominant. Confirming our findings, a previous review targeting nursing students in other areas (i.e., not mental health nursing) showed that there were a research area bias and a predominance of qualitative research [35]. In Korea, studies on the outcomes of consumer involvement have been conducted partially in the social welfare field—not in the nursing area—and this type of research is still at its initial stages [36,37]. This may be a result of specific barriers that hinder the provision of this type of education, including the need for systematic preparation [38], the lack of educational/financial support in schools [38], the lack of compensation for consumers [38], and/or ethical issues related to consumer involvement [39].

Five out of the 10 analyzed qualitative studies had more than 50 participants. These findings regarding the sample size do not concur with those of Scammell et al. [35]: Although qualitative research accounted for most reviewed research in both studies (i.e., ours and theirs), Scammell et al. faced limitations when trying to conduct in-depth analyses of the results owing to the small sample size in the reviewed studies. Additionally, the quantitative research we reviewed mainly utilized a one-group pretest–posttest design. Although this research design is useful to identify the outcomes of the educational programs when having a comparison group is not possible, we suggest that future studies should verify the outcomes of education using consumer involvement through more structured designs. This is important because the pretest–posttest design offers threats to internal validity (e.g., a maturation effect could occur among students from pre to posttest) [40].

Most studies in education using consumer involvement focused on patient recovery. Nonetheless, they were not limited to the understanding of the concept and the process of recovery; they also presented elements to help with patient recovery and promoted students’ understanding by explaining academic theory together with descriptions of consumers’/patients’ experiences. Previous studies in various fields [36,41,42] have attempted to combine knowledge related to recovery acquired through consumers’ life experiences with theoretical knowledge in student education. Our study results presented a similar context. However, we suggest that future studies should try and expand the topics of education in which the consumers partake; doing so may help to improve students’ interpersonal competency [41] and their understanding regarding the situation of the family and the care providers of the mentally ill [43].

Although consumers served as simple feedback providers or facilitators through face-to-face interactions during the educational interventions in some studies, most studies reported that the consumers contributed to the overall composition of the curriculum. Moreover, most studies remarked that such a contribution could be seen even during the planning stages of education. In particular, such contribution to the overall composition of the curriculum was often achieved through collaboration between consumers and professional training personnel (e.g., educators) [28]. Nonetheless, most reviewed studies lacked a proper description of consumer’ roles in the applied educational interventions, which could be important information when planning future programs regarding education using consumer involvement; moreover, when the studies described these roles, these were often different between the studies. Such disparate approaches to a similar type of education highlight a lack of consistency and a risk that education using consumer involvement may become tokenistic or ad-hoc methodology [44,45]. Therefore, to enhance the effectiveness of education using consumer involvement, we highlight the need for consistent and systematic management, which should follow specific standards for consumers’ roles in educational interventions. Previous studies have highlighted the need for an integrated approach that provides support for the preparation of educational interventions related to consumer involvement and that enhances consumers’ understanding of the overall purpose and process of education with consumer involvement [27,46]. Furthermore, as highlighted by the recent COVID-19 pandemic outbreak, it may be necessary to consider more flexible approaches to education using consumer involvement, namely, media other than face-to-face interventions.

The evaluation of consumer involvement in mental health nursing education was categorized into changes in students’ perceptions of education using consumer involvement, people with mental health problems, and mental health nursing education; also, changes were observed in how students experienced and assessed education using consumer involvement. These results are consistent with prior studies on the educational outcomes of education using consumer involvement for clinical experts, namely, nurses or students in the nursing area (not in the mental health nursing field) [35,47]. A study on mental health experts showed similar results to ours as the education provided to students gave them insights about the consumer’s/patient’s perspective, mental health problems, lives, and how to improve mental health services [47]. However, given the characteristics of students who take mental health nursing classes (compared to those of nursing experts), our results differ from those of the previously cited study as the educational outcomes in our setting were partly related to the development of competencies associated with becoming a mental health nurse (e.g., changes in attitudes toward mental health nursing learning, in students’ self-awareness, their reflection capacity, and empathy). Nonetheless, during the assessment of the educational intervention, the students pointed out that, regardless of the various advantages mentioned above, there is a need for a systematic implementation of planned and organized programs using consumer involvement, the development of programs that have compatible educational structures to the existing education, and the coordination and change to objective and more balanced content. This is also emphasized by Paul and Holt’s [45] report that education using consumer involvement in the mental health area should include more strategic support and organic contexts.

To summarize, based on the above mentioned achievements and improvements, we believe that the development and application of education using consumer involvement may help diversify domestic mental health nursing education methods, deepen learners’ understanding, and help develop nursing competencies related to mental health. This study provides data that can be used to develop more effective mental health nursing education programs; indeed, it presents specific results of quantitative and qualitative studies on the contents, methods, and outcomes of consumer involvement in mental health nursing education in the last 10 years. In particular, such an attempt is quite novel in Asia. Therefore, the results of this study can provide useful information in planning mental health nursing education using consumer involvement in Asia.

However, this study has some limitations. First, the extracted literature is concentrated in a specific region, so the results have limited generalizability. Second, there is a limitation in understanding the implications of education using consumer involvement that was applied before 10 years ago because the scope of analysis was limited to the period 2011–2020; thus, the study can only identify more recent trends. Therefore, further research is needed to identify the overall flow of the methods, contents, and outcomes of education using consumer involvement throughout recent decades. Third, although this study analyzed educational outcomes from students’ perspectives, it did not include the perspectives of educators or consumers participating in education. Therefore, an extensive study to comprehensively analyze these educational outcomes from multiple perspectives is warranted. Fourth, 10 out of 14 analyzed studies used qualitative methods. As more quantitative studies applying rigorous methodologies have been conducted, it is necessary to verify the educational effect of user involvement in mental nursing education.

## 6. Conclusions

The purpose of this study was to identify the main contents, methods, and outcomes of consumer involvement in mental health nursing education through an integrated literature review. Our results showed that studies on education using consumer involvement in the mental health nursing context for the past 10 years have been concentrated in certain countries and focused on a qualitative design. Additionally, the main topic of this type of education was patient recovery, and the consumers were actively involved in the educational programs ever since their planning stages. Finally, the educational outcomes were categorized into changes to students’ perceptions of education using consumer involvement, people with mental health problems, and mental health nursing education, as well as changes in students’ experiences and evaluations of education using consumer involvement. The findings of this study provide useful data that can be used as a guide for future applications of consumer involvement in the field of mental health nursing education. We believe that this study offers valid directions and information for any attempts to change mental health nursing education in North America as well as Asian countries, including South Korea.

## Figures and Tables

**Figure 1 ijerph-17-06756-f001:**
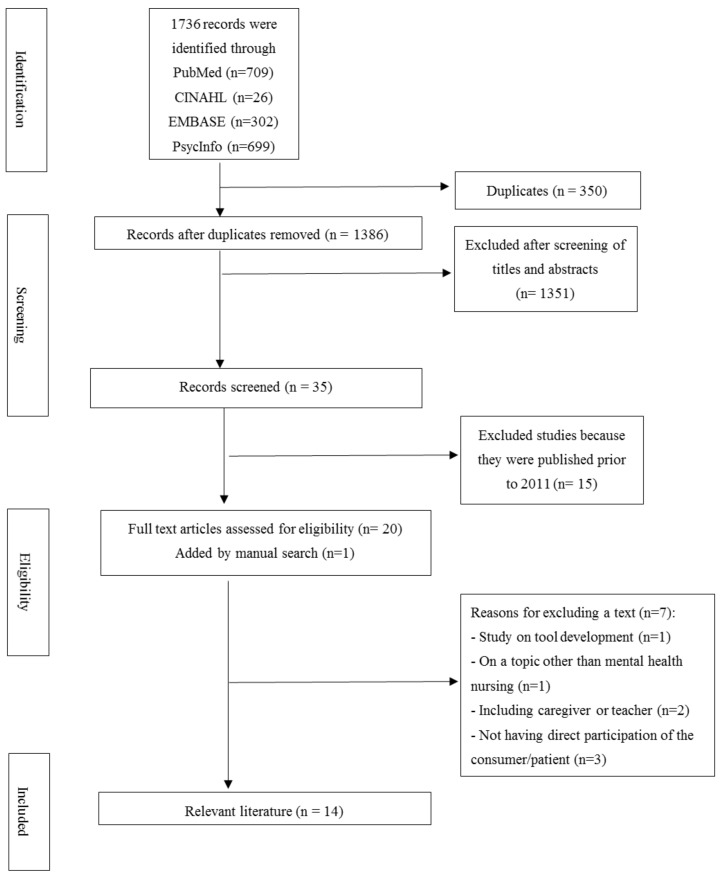
PRISMA flow chart of study selection. PRISMA = preferred reporting items for systematic review and meta-analysis.

**Table 1 ijerph-17-06756-t001:** General characteristics of the reviewed papers.

Category	Content	n	%
Country	Australia	5	35.7
UK	3	21.4
Multisite	6	42.9
Published Year	2011	1	7.1
2013	3	21.4
2014	2	14.4
2015	1	7.1
2018	1	7.1
2019	5	35.8
2020	1	7.1
Research Design	Quantitative study	4	28.6
(pre–post design)		
Qualitative study	10	71.4
(thematic analysis, phenomenological study)

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
