# Peer review of "Outcomes of Consumer Involvement in Mental Health Nursing Education: An Integrative Review"

_ijerph, 2020, doi:10.3390/ijerph17186756_

Round 1
Reviewer 1 Report
I think this manuscript is interesting. The topic is timely, and method and result sections are sound.
The idea of the review was to explore the outcomes of consumer involvement in mental health nursing education. The authors selected 14 studies from six electronic databases and summarized the results. Thereby they filled a gap for a general review for consumer involvement in mental health nursing education. The major strength is that the article is the first of its kind to give an overview of main contents, methods, and outcomes of consumer involvement in mental health nursing education. It underlines the actuality and importance of consumer involvement in mental health nursing education. It should be noted that the studies are well selected and most of the studies had a huge number of participants. The flow diagram and the selection process of the studies is reasonable.
Nevertheless, I have some further minor amendments.
- Limitations of the study are that some relevant variables such as mental health, medical education have not been included. Mental health nursing education is a complex topic because of its many different facets. This complexity was not displayed in this review.
- It should be stressed that in the literature also other terms than mental health nursing education use can be found (such as mental health nursing, etc.). A brief analysis of these terms is required.
- The focus of this study is exploring the outcomes of consumer involvement in mental health nursing education, but review of the literature in this manuscript is insufficient review of the literature of this topic. It does not put forward the possible internal mechanism of consumer involvement in mental health nursing education based on literature review. So, the inherent logic of study is not rigorous.
- Please follow the rules of English scientific writing in the manuscript. There are few types and grammatic mistakes.
In short, I personally feel that the manuscript needs to be revised.
Author Response
Thank you for this opportunity to revise our manuscript, now titled “Outcomes of Consumer Involvement in Mental Health Nursing Education: An Integrative Review.” We have carefully considered the reviewers’ comments and provided a point-by-point explanation of how we revised the paper based on the reviewers’ comments and recommendations. We hope that these revisions improve the paper such that you and the reviewers now deem it worthy of publication in International Journal of Environmental Research and Public Health.
The idea of the review was to explore the outcomes of consumer involvement in mental health nursing education. The authors selected 14 studies from six electronic databases and summarized the results. Thereby they filled a gap for a general review for consumer involvement in mental health nursing education. The major strength is that the article is the first of its kind to give an overview of main contents, methods, and outcomes of consumer involvement in mental health nursing education. It underlines the actuality and importance of consumer involvement in mental health nursing education. It should be noted that the studies are well selected and most of the studies had a huge number of participants. The flow diagram and the selection process of the studies is reasonable.
Nevertheless, I have some further minor amendments.
Point 1: 1. Limitations of the study are that some relevant variables such as mental health, medical education have not been included. Mental health nursing education is a complex topic because of its many different facets. This complexity was not displayed in this review.
Response 1: Thank you for your thoughtful comments. This study focused on attributes and outcomes of mental health nursing education programs, which used consumer involvement with undergraduate students. The authors are actually planning mental health nursing programs for undergraduate students using consumer involvement based on the results of this study. For these reasons, when searching for papers to be analyzed, only search terms related to “mental health nursing” and “education” were used, instead of others such as “mental health” or “medical education” mentioned by the reviewer. The content of the used search expression is as follows.
((mental health nursing) OR (psychiatric nursing)), ((expert by experience) OR (service user) OR (user involvement) OR (consumer participation)), and ((education) OR (training))
Point 2: It should be stressed that in the literature also other terms than mental health nursing education use can be found (such as mental health nursing, etc.). A brief analysis of these terms is required.
Response 2: Thank you for your comment. The authors would like to clarify that the terms “mental health nursing education” and “mental health nursing,” were not used as synonyms in this study. Mental health nursing is a larger concept that includes not only mental health nursing education, but also actual mental health nursing in clinical or community settings. Therefore, this study only focused on “mental health nursing education,” which refers to a more specific topic. Also, other similar terms have not been considered because these would be beyond the scope of the review in this study. As the reviewer recommended, a brief explanation of the scope of “mental health nursing education” (3.3. Literature search) in this study was described as follows.
-> The scope of mental health nursing education was based on the regular curriculums conducted at the college level for undergraduate students. (page 3, lines 20-22).
Point 3: The focus of this study is exploring the outcomes of consumer involvement in mental health nursing education, but review of the literature in this manuscript is insufficient review of the literature of this topic. It does not put forward the possible internal mechanism of consumer involvement in mental health nursing education based on literature review. So, the inherent logic of study is not rigorous.
Response 3: Thank you for your thoughtful comments. In the light of the reviewer’s concerns, the internal mechanisms related to the major outcomes of the education mentioned in the analyzed papers have been added as follows.
-> Consumers facilitated a new level of understanding for the students and created the learning process in which the students engaged, improving the quality of mental health nursing education. The education provided many opportunities to learn about mental health nursing directly from consumers who have lived experience of mental illness. (page 11, lines 6-10)
The consumer-involved education, which reflected consumers’ lives in a broader context, beyond their diagnosis, facilitated the reexamination of undergraduate students’ preconceptions and untested ideas about mental illness. (page 11, lines 15-18)
However, some students experienced negative changes, such as an excessive anxiety or discomfort caused by the desire to be recognized by a facilitator (consumer) and hide personal challenges or incompetence during class. (page 11, lines 28-30)
Point 4: Please follow the rules of English scientific writing in the manuscript. There are few types and grammatic mistakes.
Response 4: We used the services of a professional editing company and this paper was proofread by a native English speaker. Thank you.

Reviewer 2 Report
My critical comments are summarized as follows:
1. Lack of clarity on "integrated" approach.
2. Lack of clear research aims or research questions.
3. Limited patient engagement approach: the scope should be broadened from a population health management approach (patient engagement).
4. Limitations of the qualitative studies.
5. Broadening the conclusions: A global approach or perspective could be considered.
Author Response
Thank you for this opportunity to revise our manuscript, now titled “Outcomes of Consumer Involvement in Mental Health Nursing Education: An Integrative Review.” We have carefully considered the reviewers’ comments and provided a point-by-point explanation of how we revised the paper based on the reviewers’ comments and recommendations. We hope that these revisions improve the paper such that you and the reviewers now deem it worthy of publication in the International Journal of Environmental Research and Public Health
Point 1: Lack of clarity on "integrated" approach.
Response 1: Thank you for your comment. The authors followed Whittemore and Knafl’s five steps for an integrated literature review (Whittemore & Knafl, 2005). In particular, the data removal, data display, data comparison, conclusion drawing, and verification steps suggested by Whittemore and Knafl were thoroughly followed at the data analysis stage for integrated results. The authors added a more detailed explanation of the integrated approach to the manuscript as follows.
-> We followed Whittemore and Knafl’s five steps for an integrated literature review: (1) problem identification, (2) literature search, (3) data evaluation, (4) data analysis, and (5) result presentation [23]. In particular, at the data analysis stage, we thoroughly followed the data removal, data display, data comparison, conclusion drawing, and verification steps suggested by Whittemore and Knafl [23] for integrated results. (page 2, lines 43-45/ page 3, lines 1-2)
Point 2: Lack of clear research aims or research questions.
Response 2: Thank you for your comment. We specified that the aim of this study was to examine the attributes and outcomes of mental health nursing education using consumer involvement; we focused on nursing students in Korea and abroad. We clarified the aim of the study and the research questions in the text. (page 2, lines 41-43/page 3, lines 4-8)
Point 3: Limited patient engagement approach: the scope should be broadened from a population health management approach (patient engagement).
Response 3: Thank you for your comment. Since the purpose of our study was to conduct a review on patient engagement in “mental health nursing education”, the scope had to be limited.
Point 4: Limitations of the qualitative studies.
Response 4: Thank you for your comment. We have revised the manuscript by adding a limitation related to qualitative studies.
-> Fourth, 10 out of 14 analyzed studies used qualitative methods. As more quantitative studies applying rigorous methodologies have been conducted, it is necessary to verify the educational effect of user involvement in mental nursing education.
(page 13, lines 46-49).
Point 5: Broadening the conclusions: A global approach or perspective could be considered.
Response 5: Thank you for your suggestion. We broadened the conclusions adding the following consideration, “Furthermore, as highlighted by the recent COVID-19 pandemic outbreak, it may be necessary to consider more flexible approaches to education using consumer involvement, namely, medias other than face-to-face interventions.” For a global approach, We added the following sentence to the conclusion.
-> in North America as well as Asian countries, including South Korea (page 14, lines 13-14).
References
Whittemore R.; Knafl, K. The integrative review: Updated methodology. J Adv Nurs 2005, 52(5), 546-553. doi: 10.1111/j.1365-2648.2005.03621.x

Reviewer 3 Report
Revision
Manuscript: ijerph-918858
I found the manuscript very interesting and dealing with an important subject in the education of nurses specializing in mental health. The perspective of viewing mental health patients as consumers offers an opportunity for nurses to change their attitude towards a more positive light.
The lack of line numbers in the manuscript made the revision more complicated but I hope the authors will be able to find comments easily in the text.
Main concern about the manuscript
The introduction is short and concise. The study of Consumer Involvement in Mental Health Nursing Education is a new subject. Following the authors, “The utilization of the consumer involvement concept in the mental health nursing education field of research started in Melbourne, Australia, in 2001 [14], and is now the most developed field regarding consumer involvement in the context of education [15]. (first paragraph in the Background section).
When I finished my revision I asked myself: why did the authors perform a review over the last 10 years in a subject so new it’s first study was published in 2001? Why did they not include the 15 papers prior to 2011?
In the same way that the discussion highlighted the qualitative studies with more than 50 participants, the number of studies included in a revision of a subject is also an important aspect in a research.
I think it is a pity that 15 studies were dismissed only due to them being published prior to 2011. Of course the authors could prioritise the more recent papers published in the last 10 years but I believe that papers from 2001 could potentially be interesting, important, and not outdated enough to be dismissed. Basically it would just be a way of including every paper involved in a subject from the beginning having 29 papers maximum, or rather just thinking that half the papers from the last 10 years are enough.
Minor aspects
- Introduction, in the first line of the third paragraph I would include a linking word, such as “On the other hand, a consumer is a person who……”
- Background, last paragraph, third line: “experience utilization, utilized in….”. Please, find a synonym of “utilization” or “utilized” such as use or used.
- Research design, last paragraph, the sentence “After screening the titles and abstracts, 1372 documents were excluded, totaling 14 documents selected for the final analysis” should change the following aspects:
- a) totaling should be “totalling”
- b) As it can be seen in the flow chart, the authors excluded studies for the publication year and for other reasons that should be included in the text.
Author Response
Thank you for this opportunity to revise our manuscript, now titled “Outcomes of Consumer Involvement in Mental Health Nursing Education: An Integrative Review.” We have carefully considered the reviewers’ comments and provided a point-by-point explanation of how we revised the paper based on the reviewers’ comments and recommendations. We hope that these revisions improve the paper such that you and the reviewers now deem it worthy of publication in International Journal of Environmental Research and Public Health.
I found the manuscript very interesting and dealing with an important subject in the education of nurses specializing in mental health. The perspective of viewing mental health patients as consumers offers an opportunity for nurses to change their attitude towards a more positive light.
The lack of line numbers in the manuscript made the revision more complicated but I hope the authors will be able to find comments easily in the text.
Point 1: The introduction is short and concise. The study of Consumer Involvement in Mental Health Nursing Education is a new subject. Following the authors, “The utilization of the consumer involvement concept in the mental health nursing education field of research started in Melbourne, Australia, in 2001 [14], and is now the most developed field regarding consumer involvement in the context of education [15]. (first paragraph in the Background section). When I finished my revision I asked myself: why did the authors perform a review over the last 10 years in a subject so new it’s first study was published in 2001? Why did they not include the 15 papers prior to 2011?
In the same way that the discussion highlighted the qualitative studies with more than 50 participants, the number of studies included in a revision of a subject is also an important aspect in a research.
I think it is a pity that 15 studies were dismissed only due to them being published prior to 2011. Of course the authors could prioritise the more recent papers published in the last 10 years but I believe that papers from 2001 could potentially be interesting, important, and not outdated enough to be dismissed. Basically it would just be a way of including every paper involved in a subject from the beginning having 29 papers maximum, or rather just thinking that half the papers from the last 10 years are enough.
Response 1: Thank you for your comments. We are regretful too for not being able to include all the studies published on this topic. However, the purpose of our study was to review more recent research. Therefore, we decided to focus on the last 10 years. In addition, Terry (2012) provides a review on the same subject, even though it is not an integrative review.
Terry J. Service user involvement in pre‐registration mental health nurse education classroom settings: a review of the literature. Journal of Psychiatric and Mental Health Nursing, 2012, 19(9), 816-829. doi.org/10.1111/j.1365-2850.2011.01858.x
Point 2: Introduction, in the first line of the third paragraph I would include a linking word, such as “On the other hand, a consumer is a person who……”.
Response 2: Thank you for your comment. We revised the sentence with a linking phrase. Since we are not particularly drawing a contrast here, we have changed this to “It is noteworthy that… “ I hope it elucidates our meaning better.
->‘It is noteworthy that a consumer is a person who is~’ (page 1, line 40)
Point 3: Background, last paragraph, third line: “experience utilization, utilized in….”. Please, find a synonym of “utilization” or “utilized” such as use or used.
Response 3: Thank you for your comment. We revised the sentence as you suggested.
-> of consumer experience utilization implemented in nursing education differ by country. (page 2, lines 30-31)
Point 4: a) totaling should be “totalling
Response 4: Thank you for your comment. We corrected the spelling as you suggested.
-> totalling 14 documents selected~. (page 3, lines 38)
Point 5: b) As it can be seen in the flow chart, the authors excluded studies for the publication year and for other reasons that should be included in the text.
Response 5: Thank you for your comment. We described the inclusion and exclusion criteria in detail in the literature search stage. Therefore, we did not include it in the text because we thought it was redundant.

Round 2
Reviewer 2 Report
The revised version is acceptable. No further comments are listed.
Reviewer 3 Report
The manuscript is ready to be published.